

# Blockchain-enabled infrastructural security solution for serverless consortium fog and edge computing

Abdullah Ayub Khan[1,2], Asif Ali Laghari[3], Abdullah M. Baqasah[4], Roobaea Alroobaea[5], Ahmad Almadhor[6], Gabriel Avelino Sampedro[7,8] and Natalia Kryvinska[9]

[1] Department of Computer Science, Benazir Bhutto Shaheed University Lyari, Karachi, Sindh, Pakistan
[2] Department of Computer Science, Sindh Madressatul Islam University, Karachi, Sindh, Pakistan
[3] Software Collage, Shenyang Normal University, Shenyang, China
[4] Department of Information Technology, College of Computers and Information Technology, Taif University, Taif, Saudi Arabia
[5] Department of Computer Science, College of Computers and Information Technology, Taif University, Taif, Saudi Arabia
[6] Department of Computer Engineering and Networks, College of Computer and Information Sciences, Jouf University, Sakaka, Saudi Arabia
[7] Center for Computational Imaging and Visual Innovations, De La Salle University, Manila, Philippines
[8] Faculty of Information and Communication Studies, University of the Philippines Open University, Los Baños, Philippines
[9] Department of Information Management and Business Systems, Faculty of Management, Comenius University Bratislava, Bratislava, Slovakia

Corresponding authors
Abdullah Ayub Khan,
abdullah.ayub@bbsul.edu.pk
Asif Ali Laghari,
asiflaghari@synu.edu.pk

## ABSTRACT

The robust development of the blockchain distributed ledger, the Internet of Things (IoT), and fog computing-enabled connected devices and nodes has changed our lifestyle nowadays. Due to this, the increased rate of device sales and utilization increases the demand for edge computing technology with collaborative procedures. However, there is a well-established paradigm designed to optimize various distinct quality-of-service requirements, including bandwidth, latency, transmission power, delay, duty cycle, throughput, response, and edge sense, and bring computation and data storage closer to the devices and edges, along with ledger security and privacy during transmission. In this article, we present a systematic review of blockchain Hyperledger enabling fog and edge computing, which integrates as an outsourcing computation over the serverless consortium network environment. The main objective of this article is to classify recently published articles and survey reports on the current status in the domain of edge distributed computing and outsourcing computation, such as fog and edge. In addition, we proposed a blockchain-Hyperledger Sawtooth-enabled serverless edge-based distributed outsourcing computation architecture. This theoretical architecture-based solution delivers robust data security in terms of integrity, transparency, provenance, and privacy-protected preservation in the immutable storage to store the outsourcing computational ledgers. This article also highlights the changes between the proposed taxonomy and the current system based on distinct parameters, such as system security and privacy. Finally, a few open research issues and limitations with promising future directions are listed for future research work.

# INTRODUCTION

In the past few years, fog computing has emerged as an effective computing technology to play a vital role in outsourcing computation and fulfilling the growing demand of connected stakeholders to process their requests using fog data nodes (*Mahmud, Ramamohanarao & Buyya, 2020*; *Khan et al., 2022*). Nowadays, the increased demand and usage of Internet of Things (IoT)-enabled connected multimedia devices and sensor network-based applications, such as ubiquitous and sensory devices, have escalated at a rapid rate. Due to this, the nodes of fog-enabling technology connect all these devices on a single platform. The need to reduce the rush over the transmission channel is due to handling optimization-related issues and challenges, such as key parameters of quality of service, including latency, bandwidth, transmission power, delay, throughput, response, duty cycle, privacy and security, and efficient computation. The paradigm of fog computing tackles all the emerging barriers in the IoT environment; most probably, the biggest issue is security for all the fogs and IoT devices that connect, communicate, and interact efficiently (*Zahmatkesh & Al-Turjman, 2020*; *Khan et al., 2021*). These challenges and issues are addressed by different proposed research projects in which the concept of fog computing and its paradigm with IoT technology are utilized in academia and industry to create an effective distributed, connected environment.

The recent development of fog computing and its adaptation as an outsourcing method are going to make this a more attractive research area. The technology gets several inputs in terms of review articles, tutorials, short opinions, and surveys that have been issued in the last few years (*AlBadri, 2022*; *Sarker et al., 2022*; *Lei et al., 2020*). There is various literature related to fog management that has been reported (along with data security and privacy-related issues) with several independent architectures, and models of fog virtualization and computation integration with cryptographic encryption are proposed, as shown in Fig. 1.

These related infrastructures handle node transactions securely. Most of the architectures are proposed by examining and analyzing different edge-based distributed applications (*Lin et al., 2020*). However, IoT-enabled devices and the edge change the scenario of distributed computing topology, which processes information that is located near the edge. Through this process, IoT-enabled systems and end-users produce as well as consume the information. The purpose is to bring computation and data storage closer to the devices where the data is being gathered. It cannot rely on the central location, which can be hundreds of miles away. In a real-time environment, the transmission of data does not suffer in terms of latency, delay, duty cycle, or response-related issues, which directly affect the performance of distributed applications. It can reduce the cost and increase productivity by having the processing done locally (*Caiza et al., 2020*). It can also reduce

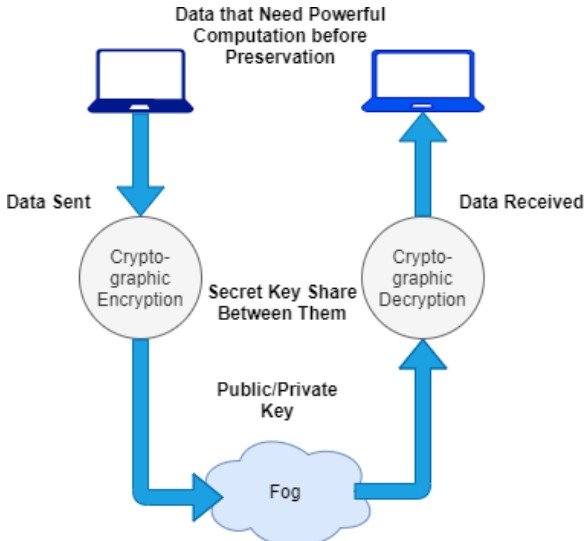

**Figure 1** **The existing scenario of fog-enabled outsourcing computation.**

the amount of data that needs task scheduling, processing, and management in a fog node environment.

The exponential growth of Internet of Things (IoT)-enabled devices and their connectivity on the network creates security and privacy-related challenges when transmitting information from one end, receiving it from the other, and delivering it back to the fog node (*Chalapathi et al., 2021*). However, many IoT-enabled devices generate a huge amount of data during the processing of events. This additional load of data scheduling, processing, managing, and organizing in the fog nodes consumes more time and costs more to preserve all the processed information in the distributed node environment. These pose a serious issue when creating the fog nodes as an outsourcing computation integrated with different IoT devices (*Tange et al., 2020*; *Ahmed et al., 2021*). The security of the data transmission channel while exchanging information between stakeholders and preserving that information in the storage required raises concerns in terms of protection. For this purpose, to ensure the validity, authenticity, and reliability of fog-based computation in a distributed environment, it is imperative and most significant to maintain the integrity and transparency of the complete process.

Blockchain distributed ledger technology is enabling organizations to protect their current infrastructures and protocols, specifically edge-related transactions, to realize integrity, privacy, traceability, provenance, and direct access to information *via* distributed applications (*Li et al., 2021*). It also accesses the application programming interface in every aspect of information technology to protect the events of node transactions, especially edge-based transactions. In a fog-based integrated edge environment, blockchain provides secure, encrypted, and protected preservation of processed information and related transactions in the immutable ledger to enable the transparency of the computational process of the designed infrastructure. However, the edge-based chain of transactions and events

of nodes information is preserved in chronological order in the chain structure, which connects different channels in the consortium architecture of a blockchain Hyperledger network. In addition, the smart contract is allowed to design and create a platform to manage, control, and achieve the distributed autonomous computation application in an outsourcing fog environment. By this act, the system receives a secure, transparent, and immutable IoT-enabled edge-based generated ledger, which is hard to tamper with and forge because of NuCypher threshold proxy re-encryption protection and stores all these ledgers in a block-chain-fog-enabled secure distributed preservation container (*Nazir et al., 2022*; *Al-Turjman, Zahmatkesh & Tariq, 2021*).

However, many cloud computing experts are adopting blockchain Hyperledger technology. In fact, researchers are shifting towards the decentralized distributed environment, which provides a modular architecture that protects against various malicious threats usually intended for the existing centralized server-based infrastructures (*Bera & Misra, 2020*). Blockchain distributed ledger technology enables outsourcing computation environments to robust node defense capabilities with the help of hashing functions and NuCypher re-encryption deployment for intrusion detection and the secure entry of processed information in the storage (*Sarker et al., 2021*; *Zhang et al., 2021*). Further, it provides a platform to install firewalls, anti-disclosure techniques, and procedures to guarantee ledger integrity, transparency, provenance, immutability, and trustworthiness within the node and explicitly transmitted channel environments.

Substantially, it also reports that there are various related surveys and review articles published, which are unable to introduce the taxonomy of blockchain Hyperledger-enabled architectures for edge computing or integrate the out-sourcing nodes for computation specifically. In fact, there is a need to create an efficient procedure for the events of node transactions in fog-based computational architectures to analyze the existing status of the research and examine the different research problems in the literature. This research presents a systematic review of the current work by considering the architectures and models of secure distributed edge computing and the procedure to connect outsourcing nodes for efficient computation.

In this systematic review, we studied and investigated various related papers in accordance with edge-based data collection and computation. After deep analysis, there is a big research gap that needs serious concern. In this manner, a novel and secure edge computing taxonomy is designed that is integrated with fog node-enabled computations and preserves all the processed information using blockchain distributed ledger infrastructure for the sake of privacy and security. The proposed architecture provides information integrity, provenance, traceability, and assurance of service delivery for distinct operations in a serverless chain-like structure. The events of operations are demonstrated in the following order: (i) capturing edge-based records, (ii) scheduling, (iii) computing, (iv) managing, and (v) organizing individual entities on the fog nodes, and (vi) preserving validated records in the blockchain's distributed immutable storage and interpreting the information among participating stakeholders (if required). This proposed taxonomy ensures the privacy and protection of the overall node transactions in distributed stored information using a NuCypher threshold proxy re-encryption mechanism. However,

**Table 1  Acronym description.**

| Acronym | Explanation |
| --- | --- |
| IoT | Internet of Things |
| NuCypher | NuCypher Threshold Proxy Re-Encryption |
| E-Healthcare | Electronic Healthcare |
| IT | Information Technology |
| P2P Network | Peer to Peer Network |
| CA | Certificate Authority |
| IPFS | InterPlanetary File Storage |
| DApp | Distributed Application |
| DDoS | Distributed Denial of Service |

Table 1 expresses the description of acronym uses in this systematic review article as follows.

## Research motivation, objectives, and contributions

In this article, we highlight the main objectives, which address efficient computation, security, and privacy-related issues. It includes the characteristics of edge computing, the role of blockchain technology to protect ledger transactions, and the importance of outsourcing computation in the edge/IoT-based complex environment. In addition, we identify various related limitations in the architecture design, range of parameters of quality of services, development and deployment details, transmission and communication protocols, and applicational modes. However, the proposed architecture for edge computational integration with outsourcing nodes is based on the current central systems and compared with different state-of-the-art methods based on taxonomy. We present the contents of further improvement, performance enhancement, opportunities, and futuristic implementation details to create and deploy an efficient architecture for edge-based distributed computing in a secure manner.

The major contributions of this systematic review are mentioned as follows:

- In this article, we study various fog computing, edge computing, and blockchain-enabled architectures and their transactional details. Therefore, we conducted a systematic review of secure edge-based integrated outsourcing computation.
- Identify the existing issues and current status of edge computing research and analyze a few research problems involved in different relevant domains, such as edge security and privacy.
- The blockchain Hyperledger-enabled secure distributed architecture is proposed for edge-IoT-related computations integrated with outsourcing nodes and preserving information in immutable storage according to the designed taxonomy.
- Compared the existing studies with the proposed architecture and examined the designed taxonomy based on a different range of parameters accordingly. It helps to identify and analyze edge-related protected categories.
- Finally, we identify, examine, and evaluate the key challenges and limitations involved in this systematic review of distributed edge computing, especially in the security and

privacy domains. Highlighted and discussed the few open research challenges and the possible solutions, along with the future direction.

The remainder of this systematic review is organized as follows: In 'Survey on Edge Computing', edge computing and enabling technologies are discussed in the context of systematic adaptation and analysis. The use of fog nodes as an outsourcing computing and preservation technology, the related research questions, and their possible solutions are discussed in 'Fog computing as outsourcing computing technology'. 'Solutions of different research questions' and 'Data security in edge-based outsourcing computation and cloud preservation' are oriented towards the security and privacy of information in edge outsourcing node-enabled computations and the role of blockchain Hyperledger technology to protect the integrity and transparency of processed ledgers, transmission channels, connectivity, and the network environment. In 'Data security in edge-based outsourcing computation and cloud preservation', we propose a secure and consortium architecture for edge computing processes that are integrated with outsourcing computation using blockchain Hyperledger Sawtooth. 'Current State-of-the-Arts' highlights open research issues, which are the major portions that need concern, and future objectives with research directions are discussed. Finally, this systematic review concludes in 'Conclusions'.

## SURVEY ON EDGE COMPUTING

With the increasing number of IoT-enabled multimedia devices, sensory networks, ubiquitous connectivity, mobile internet, and other different objects connected over the network, this generates a massive amount of data. According to the survey (*Wei, Yang & Wang, 2020*), almost 70 billion devices will connect to the network over the coming years of 2022–2023, which would generate approximately twice as much data as 2020–2021. Therefore, it is hard to handle such data and compute individually using the current computational models for examining, analyzing, managing, and optimizing, such as cloud-based client–server and decentralized distributed computing. For applicational design, it is necessary to manage fast response and supportive mobile activities, for example, autonomous response in a smart industrial environment, clinical and emergency care in healthcare, intelligent monitoring, and distributed power systems (*Choudhury et al., 2021*). In order to control the challenges involved in reducing complexity, they are as follows: (i) high latency and bandwidth issues; (ii) privacy and security sensitivity; and (iii) dispersion in geography. In this scenario, there is a need for an efficient computing paradigm that assists fog computing, edge computing, and related processes with the aim of connecting IoT-enabled multimedia devices with a minimum amount of latency and response time (*Luo, Li & Chen, 2021*).

The advent of edge computing provides a new way to transform the IoT-based generated data in a well-handled, processed, and delivered manner, where billions of multimedia devices are connected directly throughout the globe, as shown in Fig. 2. This exponential growth of the internet is due to the increased number of Internet of Things devices (*Laghari, Li & Chen, 2021*). The vast connection of IoT nodes with each other for the purpose of

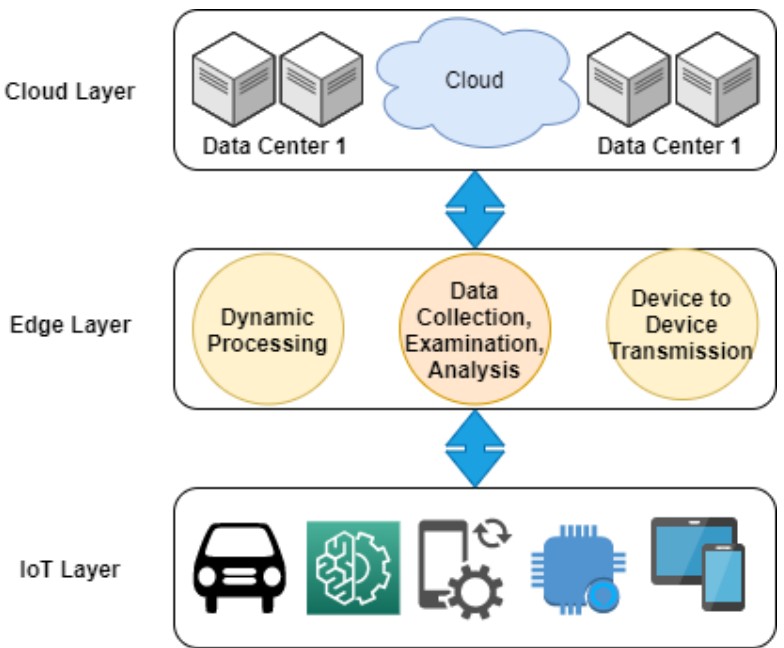

**Figure 2** Systematic review guideline for distributed and edge-enabling technology.

communicating, either receiving information from the fog or sending data points to the fog. However, various connected IoT devices generate large numbers of records during the transmission, where enormous records transactions are exchanged between a node in which a number of operations are executed in a single stream, which poses a serious issue in terms of consumption of edge resource constraints, such as computational power, network bandwidth, and storage (*Zhang, Li & Chen, 2021c*; *Liu, Rahman & Hossain, 2021*).

In an industrial environment, monitoring of industrial, manufacturing, and production units needs various internet-enabled ubiquitous devices that connect factory floors, video cameras, and other monitoring equipment for the sake of gathering live footage, data records, and supply-chain analytics from a remote office. In this manner, a problem arises when these connected nodes transmit data in the same slice (*Rahman & Hossain, 2021*). Instead, for manufacturing monitoring, multiple video surveillance nodes are connected by thousands or more, which not only affects the quality but also suffers network connectivity due to latency, delay, and variable response, which directly impact the computational cost as well. Most of these problems get solved by tuning edge computing hardware and services by managing a local source of processing units along with categories of information preservation (static and dynamic) for many of the connected devices. Further, the edge gateway allows data to be processed from the edge devices, and then it sends only the crucial or relevant processed information to the cloud storage. This complete process reduces the load on bandwidth and the additional cost of computation. Moreover, the technology involves many different things, such as wireless sensor networks, ubiquitous computing, mobile devices, surveillance cameras, and internet-enabled microwave-based

controllers for smooth transmission and delivery. However, in this scenario, the edge gateways consider edge devices within the infrastructure of edge computing.

This section discusses the background context of edge-enabling distributed computing, related surveys, and systematic observations.

## Context and analysis

Due to the increased use of IoT, edge computing gained popularity after the advent of ubiquitous healthcare, which directly impacts the medical industry in a positive manner. The use of electronic healthcare (E-healthcare) systems provides dynamic control of data transmission, effective medical services, and a fast process of service delivery from edge to edge (*Rahman & Hossain, 2021*). These bulk transactions need to be processed on the same side; to cut down on this burden on terminals, end-users outsource and record all the details to the cloud provider and preserve them in the cloud storage. However, the cloud servers are not trusted by the end-users or the owner of the data because of weak security and privacy between participating stakeholders and cloud components. Therefore, it is also because the e-healthcare data is directly associated with an individual user, which plays a vital role in diagnostics, treatment, and managing medical servers (*Abdellatif, Li & Chen, 2021*).

As we discussed earlier, 70 billion IoT devices will connect from the end of 2022 until 2023, which leads to big data-related issues. To examine these scenarios, it seems clear that it is difficult to handle a large number of connected nodes and process their generated data directly in the run-time environment. For instance, the traditional model of data processing and computation is unable to tackle these issues while using cloud and distributed computing mechanisms (*Lv et al., 2021*). Recently, in most cases, the data required quick response and mobility-related support. Therefore, in order to maintain data processing, high internet bandwidth, ultra-high latency, space for dispersing computational units, and privacy-sensitive desktop applications are required. Further, there is a need for an assistant that handles data from the IoT devices, processes it in the cloud, and preserves it.

The concept of edge computing involves devising a scenario in which user data is processed on the periphery of the system network, which is possibly near the original source (*Lahkani et al., 2020*; *Afonasova et al., 2019*), as shown in Fig. 2. In the complete scenario, it matters that the location of data is retained, moved, and processed. For instance, as we know, data is generated at an end-user node, and if it requires a process, then the system's processor traditionally executes an individual transaction. Recently, after the advent of cloud computing, the processing of data required a pay-per-use cloud scenario to execute transactions in accordance with the defined procedure. The movement of data depends on the internet, most likely a wide area network, where a local area network is used to transmit corporate or enterprise transactions. The data is preserved and accessed through the application, and the result is sent back to the end-user devices, respectively.

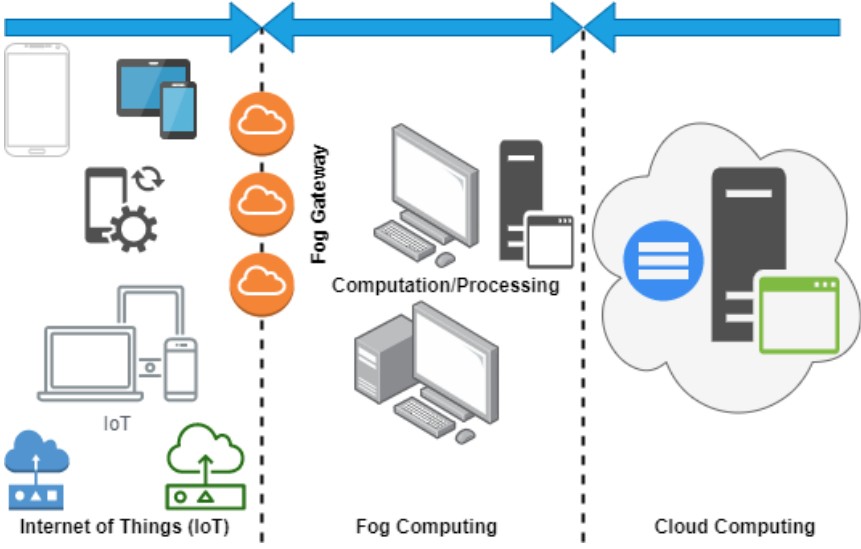

**Figure 3  Fog computing architecture.**

## Search methodology

In this section, we highlight a few works of literature related to the current trend of edge computing involving security and privacy concerns, a review of edge with cloud, and the evaluation of blockchain and its enabling technologies with edge computing (see Table 1). Further, we presented a category of paper distribution in accordance with the edge computing survey, review, and current trends and challenges and discussed accordingly, as shown in Fig. 3. However, the comparison table provides a complete understanding of previously published surveys and reviews with the proposed state-of-the-art study (see Table 2). However, the constraint of evaluation is mentioned as follows: (i) paper title, (ii) paper description, (iii) findings, and (iv) existing limitations.

## FOG COMPUTING AS OUTSOURCING COMPUTING TECHNOLOGY

The concept of fog computing was proposed by Cisco in early 2014; with the advent of this, IoT technology and related applications became more focused and utilized, which provided efficient and fast execution and delivery as compared to the traditional cloud computing models (_Dastjerdi & Buyya, 2016_; _Lis, 2021_).

Recently, there have been several white papers presented by Open-Fog to clarify the role, features, and advancements of fog computing. Location is the key factor as the technology defines the horizontal architecture that allocates resources, such as storage, network bandwidth, and computational controls, to the nearest connected nodes (_Ravindran, 2019_; _Chaveesuk, Khalid & Chaiyasoonthorn, 2021_). This technology works as a bridge between the Internet of Things and the cloud for the purpose of enhancing data collection, management, organization, optimization, computation, network transmission and delivery, and storage. Simply put, it reduces the load of traditional

**Table 2  EDGE-related literature and related analysis.**

| Paper title | Paper description | Findings | Existing limitations |
|---|---|---|---|
| Survey on Intelligence Edge Computing in 6G (*Al-Ansi, Li & Chen, 2021*) | The author of this paper presented a review report on the edge-enabling intelligent system connected to a 5G network, and so, highlight the challenging issues while moving towards 6G technology. | • Survey from 2014 to 2021<br>• This paper highlighted the key factors for the 6G network, such as architecture and the future market | • In futuristic 6G network, holographic communication is the most challenging issue<br>• Multi-sensing networks<br>• Cross-platform<br>• Time engineering platform<br>• Effective infrastructure management |
| A secure data storage and an efficient data exchanging approach for blockchain-enabled mobile edge computing (*Zhang, Li & Chen, 2021b*) | In this paper, the authors construct a unique private key regional system that is shared in multiple forms. This proposed scheme collaborates with the blockchain distributed ledger technology to provide a secure mobile-based edge-enabling data exchange, storage, and management facility. | • Designed trusted edge nodes along with the storage management system<br>• Implemented proxy server<br>• Digital signature and private key sharing strategy are presented | • Attacks on security protocols, such as cloud storage servers<br>• Computational cost of private key generation and exchange<br>• Signature time limitation<br>• Size of the signature issue |
| A recent advancement in edge-enabling artificial intelligence of things computation (*Chang, Li & Chen, 2021*) | This paper conducted an extensive survey on edge-enabling orchestrated architectural computation and analysis to find the technological role in the artificial intelligence of things environment. Further, in this paper, the authors separated the list of emerging issues, challenges, limitations, and futuristic open research domains associated with this field. | • Presented a practical artificial intelligence of things<br>• Illustrated the role of edge computing and artificial intelligence in the Internet of Things environment | • In artificial intelligence of things, the multitude of wireless sensors creates a challenging prospect in the run-time environment<br>• While collective integration of artificial intelligence and edge computing to manage large amounts of data handling posed serious complexity because of network infrastructure |
| A review on multi-access edge computing technology (*Ali, Gregory & Li, 2021*) | This review paper aimed to examine, analyze, and present the closer proximities of multi-access edge computing, such as computation, storage, and network bandwidth, to end-users. In addition, the paper provides a case study, guidelines, conceptual aspects, security concerns, and related architectures for multi-access edge computing. | • The author presented the investigational report related to the multi-access edge computing architecture and their functional layer hierarchy, along with the identified list of threads and security gaps<br>• Comprehensive perspective related to multi-access edge computation | • Security dimension and recommendation X.805<br>• End-to-end security concerns<br>• Access control of edge IV-B1-2<br>• Identification and authentication issues in heterogeneous environments |
| An edge computing-enabled cluster-based algorithm design for internet of things (*Zhang, Li & Chen, 2021a*) | The author of this paper presented a new paradigm of edge computing by enabling cluster-based IoT to manage node energy, transactional speed, routing, deliverance, and optimization. It also handles IoT-to-IoT communication modes and interoperability. | • Device-to-device mode is designed for direct communication<br>• The proposed clustering algorithm provides a reduced consumption of network bandwidth because of tuned network topology and related control overhead | • Limitation in device-to-device resource consumption<br>• Scope of data security and privacy<br>• Streamline data execution and automation |

**Table 2** (*continued*)

| Paper title | Paper description | Findings | Existing limitations |
|---|---|---|---|
| Resource allocation and management of edge computing in IoT environment (*Xu, Li & Chen, 2021*) | This paper discussed the resource trading process of edge computing for IoT devices to protect critical information in terms of ledger security and privacy. | ● Designed edge computing stations<br>● Implement a resource trading scheme<br>● Use a blockchain-based distributed network for intercommunication and connectivity | ● Platform interoperability issue<br>● Improvements are required in blockchain cross-chaining solutions<br>● Hash-Re-encryption<br>● Edge-enabling wireless sensor-based interconnectivity issue |
| A role of federated learning in edge-computing (*Xia, Li & Chen, 2021*) | In this survey paper, the author provides a new paradigm of distributed application along with the role of federated learning in an edge computing environment. Further, this paper highlighted the most suitable development tools for designing and creating distributed applications for edge-enabling technological communication structures using machine learning, along with risk mitigation procedures. | ● This proposed approach reduces the training cost<br>● Less consumption of inference time<br>● Multiple devices are handled concurrently | ● Cross-silo edge issue<br>● Intercommunication limitations in edge-to-edge<br>● Network quantization<br>● As outsourced computation occurs, there are security and privacy issues while data is traveling |

cloud-enabling data centers (*Chatterjee, Priyadarshini & Le, 2019*; *Khan et al., 2022c*) and works on decentralized computing to enhance computational executions. Currently, the technology is performing a vital role in various domains of computing, health, the environment, social science, business, enterprises, *etc.*

Figure 4 shows the complete hierarchy of fog computing along with the connectivity of cloud computing and IoT. This layered-based infrastructure is categorized into three main layers, such as (i) the IoT layer, (ii) the fog layer, and (iii) the cloud layer. However, there are various advancement reports in the industrial, distributed core functions, other supporting technologies, academic research, and others (*Yousefpour et al., 2019*; *Atlam, Walters & Wills, 2018*; *Ghobaei-Arani, Souri & Rahmanian, 2019*); also, the previous study listed the emerging and involving challenges, limitations, and issues in these domains and directly addressed the concerns of experts for the sake of technological maturity. The list of problems is discussed as follows (*Ghobaei-Arani, Souri & Rahmanian, 2019*; *Iorga et al., 2018*; *Naha et al., 2018*):

- Vulnerability, attacks, and weak security
- Data integrity and availability
- Authentication
- Ledger protection and privacy
- Operational cost
- Data management, organization, and optimization

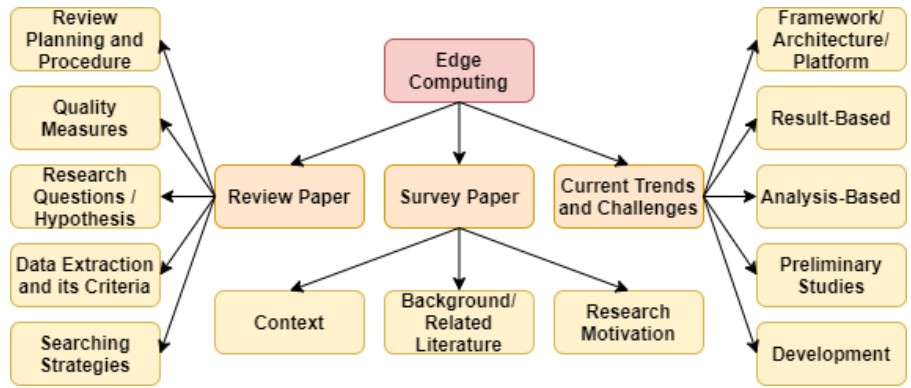

**Figure 4** **The integrated architecture of edge, fog, and cloud and the role of blockchain Hyperledger.**

## Review planning on previous studies

To initiate a review on any specific topic that is completely based on the designed rules of critical review by scheduling the research questions. This goes to the further domain of searching, where the related questions that are asked are retrieved or extracted from various connected databases (such as Science Direct, IEEE Xplore, the ACM Digital Library, *etc.*) (*Mukherjee, Shu & Wang, 2018*; *Mutlag et al., 2019*). The critical review procedure is used to identify records that are exact or close enough to detect and capture sufficient records for the requested study or research question. Therefore, it is noted that the process of review has played a prominent role in the article in determining whether it should be considered or not for further analysis (*Mutlag et al., 2019*; *Kumari et al., 2018*). All these processes reduce the risk of article selection and biases in the decision to choose a single expert's research. Therefore, in this systematic review, we have divided the tasks of critical review and selection of research questions and their possible solutions. However, all the authors in this research have prepared a list of data and shared the report with the peer authors of this article. This process has been repeated until we have all the possible research questions from the last ten years of related papers. In the whole scenario, an extensive search has been performed on the EI, SCIE, Scopus, and other related databases. The hierarchy of searching is discussed as follows:

- Search keyword definition
- Search status
- Exclusion titles
- Abstract and conclusion exclude
- Eliminate full text or extra details
- Discard common limitations, issues, and challenges
- Outline research

### Research questions and rationale plan for gathering source of information

The purpose of this study is to facilitate the researchers in terms of providing a systematic review, where they will get knowledge of the current status of the technology and the open issues and elaborate on the new prospects or remaining domains. The research questions are designed in a way that requires a process of research planning (*Zhang, Zhou & Fortino, 2018*; *Abdulkareem et al., 2019*). Tables 3 and 4 shows the research methods and case study-related research questions, their descriptions, current challenges and limitations, and future perspectives. Therefore, it is shown how the proposed systematic review is targeting the specific areas (trending areas) of fog computing and their related research questions (*Abdulkareem et al., 2019*; *Habibi, 2020*).

Recently, fog computing has been widely used as an outsourced computation. The architecture of outsourcing-enabled computation is different as compared to the existing design of fog computing, which became a new subject area of research because very few research articles were published from 2013 to 2021 (*Puliafito et al., 2019*; *Aazam, Zeadally & Harras, 2018*; *Li, Ota & Dong, 2018*). In this regard, this study highlights gaps (see Tables 3 and 4), so that the experts in the technology can choose research topics in accordance with their interests and expertise. The criteria for searching and processing are the same as those we already discussed in the above section (review planning).

## SOLUTIONS OF DIFFERENT RESEARCH QUESTIONS

In this context, we illustrate a tabular structure of previous solutions, which are presented by experts to improve the use of integrated technologies, such as Cisco and fog, fog enabling IoT, cloud/fog computational perspectives in IoT data handling, and federated learning in fog. The state-of-the-art proposed works are discussed in Tables 3 and 4. However, the constraints of the discussion of both the tables (such as Tables 3 and 4) are based on previous results that indicates to the feature improvement in the future. The list of major aspects are as follows:

- Development process
- Architecture/architecture
- Method of connectivity and intercommunication
- Real-time applications
- Hardware software
- Security and privacy
- Preservation
- Mode of transaction execution
- Control operation
- Quality-of-service (QoS) and quality-of-experience (QoE)
- Open research issues/ futuristics implementation
- Blockchain-enabled chain codes
- Network/path development
- Platform interoperability

**Table 3  Comparative analysis based on current development in edge/fog computing and related features.**

| Categories | Years | | | | | | | | | | Proposed systematic review |
|---|---|---|---|---|---|---|---|---|---|---|---|
| | 2013 | 2014 | 2015 | 2016 | 2017 | 2018 | 2019 | 2020 | 2021 | 2022–2023 | 2023 |
| | References | | | | | | | | | | |
| | *Chandra, Weissman & Heintz (2013)* | *Singh (2021)* | *Lopez, Li & Chen (2015)* | *Shi, Li & Chen (2016)* | *Satyanarayanan (2017)* | *Sonmez, Oz-govde & Ersoy (2018)* | *Khan et al. (2019)* | *Cao et al. (2020)* | *Al-Ansi, Li & Chen (2021), Chang, Li & Chen (2021)* | *Nain, Pat-tanaik & Sharma (2022)* | |
| Development process | ✓ | | | | ✓ | | ✓ | | ✓ | ✓ | ✓ |
| Architecture/architecture | | ✓ | | ✓ | | ✓ | | ✓ | | | ✓ |
| Method of connectivity and intercommu-nication | ✓ | ✓ | | ✓ | | | ✓ | ✓ | ✓ | | ✓ |
| Real-time applications | | | ✓ | ✓ | | ✓ | ✓ | | | ✓ | ✓ |
| Hardware Software | ✓ | | ✓ | ✓ | | ✓ | | ✓ | | | ✓ |
| Security and privacy | | ✓ | | | ✓ | | ✓ | ✓ | ✓ | ✓ | ✓ |
| Preservation | | | | | | | | | ✓ | ✓ | ✓ |
| Mode of Transaction execution | | | ✓ | ✓ | | ✓ | ✓ | | ✓ | | ✓ |
| Control Operation | | ✓ | | | ✓ | | | ✓ | | ✓ | ✓ |
| Quality-of-service (QoS) and Quality-of-Experience (QoE) | | | ✓ | | ✓ | | ✓ | | ✓ | | ✓ |
| Open research issues/ futuristics implementation | ✓ | ✓ | | ✓ | | ✓ | | ✓ | ✓ | ✓ | ✓ |
| Blockchain-enabled chain codes | | | | | | | | | | ✓ | ✓ |
| Network/path development | | ✓ | ✓ | | ✓ | ✓ | | ✓ | | ✓ | ✓ |

- Technological collaborative features
- Consortium network
- Public chain connectivity
- Intercommunication and associativity

**Table 4  Comparative analysis based on edge/fog computing features with blockchain integration.**

| Categories | Years | | | | | | | | | | | Proposed systematic review |
|---|---|---|---|---|---|---|---|---|---|---|---|---|
| | 2013 | 2014 | 2015 | 2016 | 2017 | 2018 | 2019 | 2020 | 2021 | 2022–2023 | | 2023 |
| | References | | | | | | | | | | | |
| | Chandra, Weissman & Heintz (2013) | Singh (2021) | Lopez, Li & Chen (2015) | Shi, Li & Chen (2016) | Satyanarayanan (2017) | Sonmez, Ozgovde & Ersoy (2018) | Khan et al. (2019) | Cao et al. (2020) | Al-Ansi, Li & Chen (2021), Chang, Li & Chen (2021) | Nain, Pattanaik & Sharma (2022) | | |
| Platform interoperability | ✓ | | | | ✓ | | ✓ | | ✓ | ✓ | | ✓ |
| Technological collaborative features | | ✓ | | ✓ | | ✓ | | ✓ | | | | ✓ |
| Consortium network | ✓ | ✓ | | ✓ | | | ✓ | ✓ | ✓ | | | ✓ |
| Public chain connectivity | | | ✓ | ✓ | | ✓ | ✓ | | | ✓ | | ✓ |
| Intercommunication and associativity | ✓ | | ✓ | ✓ | | ✓ | | ✓ | | | | ✓ |
| Processor of transactions | | ✓ | | | ✓ | | ✓ | ✓ | ✓ | ✓ | | ✓ |
| Scheduling processing | | | | | | | | | ✓ | ✓ | | ✓ |
| Anonymous/Automation | | | ✓ | ✓ | | ✓ | ✓ | | ✓ | | | ✓ |
| QoE/QoS | | ✓ | ✓ | | ✓ | ✓ | | ✓ | | ✓ | | ✓ |

- Processor of transactions
- Scheduling processing
- Anonymous/automation
- QoE/QoS

# DATA SECURITY IN EDGE-BASED OUTSOURCING COMPUTATION AND CLOUD PRESERVATION

Edge computing is the technology that deploys computational, network, and storage resources outside the data center. This infrastructure of the edge provides close-to-the-point activities, where it gets computational support from a series of connected devices, such as Internet of Things components linked to the edge device between end-users and distributed applications (*Khan et al., 2022b*; *Ranaweera, Jurcut & Liyanage, 2021*). Until now, most businesses and giant enterprises have adopted edge computing technology because of its effective structure as compared to other outsourcing models in the IT

paradigm (*Khan, Shaikh & Laghari, 2022*; *Mukherjee et al., 2020*). Edge manages machine-to-machine transactions, which lack human oversight (human intervention). Thus, edge computing-enabled data security and privacy are particularly serious issues. For instance, understanding individual prospects and turning them into remedies is one of the critical aspects necessary to ensure the secure delivery of business or enterprise operations.

However, no standard process for edge computing security has been published previously that handles secure edge device accessibility, both physical and logical interfaces for end-users, and transaction initiation to deliverance. Recently, with all this lack of standardization, edge security has become impossible because at least some control over physical access to edge components is required. It includes the accessibility of both the local access interface and edge devices together, as well as a secure portal for data capture, scheduling, organizing, and optimizing (*Shaikh & et al, 2022a*; *Zhang et al., 2018*; *Khan et al., 2022a*; *Mendki, 2019*; *Yang, Lu & Wu, 2018*). The main focus of this study is to recognize the level of security provided by the technology in the current scenario and present the best edge-to-edge physical security that falls under edge-to-edge. Therefore, we identify the security and privacy protocols used in data center technology and highlight which factors make the technology different from others.

- Data preservation, backup, protection, and privacy
- Password-based authentication
- Perimeter defense
- Cloud-enabled processing and fog-based processing
- Node-to-node (IoT) connectivity and intercommunication security

## Edge integrated with cyber security and the role of blockchain Hyperledger technology

The number of connected edge devices means increased use of potential edge gateways that become the way of system attacks. The malicious attackers are spoilt for choice when choosing the least protected (unsecure) gateways. That is the reason why edge devices get easily compromised. According to the survey reported by the University of Maryland, malicious attackers attack more than 2,000 people per day (*Yuan et al., 2021*; *Ma et al., 2019*; *Chithaluru et al., 2024*; *Gill et al., 2024*; *Asaithambi et al., 2024*; *Miao et al., 2024*; *Ali, Li & Yousafzai, 2024*; *Nandhakumar et al., 2024*; *Silitonga et al., 2024*; *Wu et al., 2024*). It is worth noting that the management of data protection is consumed more financial cost substantially. Throughout the last year and recent years, the amount spent on cyber security has been more than 130 billion dollars. However, there is an underlying limitation of cyber security and related peripherals that requires attention at each level of digital delivery. On the other side, edge computing operates within the structure of a distributed environment. The technology integrated with the internet creates vulnerabilities twice as much as in other domains, whereas edge-based connected objects are considered the weakest link and probably the most obvious gateway that can easily be hacked. It is enough to get entrance into the distribution chain of the connected node environments.

In the domain of the edge, the infiltration of the microdata centers is only possible through hardware or software (such as manipulation), and so through the classical types

of attacks such as distributed denial of service (DDoS). To solve such issues in distributing real-time, an approach of risk mitigation is applied, namely Security by Design, which encounters terms of minimizing risk and maintaining the privacy of the transactions (*Ali, Li & Yousafzai, 2024*; *Nandhakumar et al., 2024*; *Silitonga et al., 2024*; *Wu et al., 2024*; *Gill et al., 2024*). The primary task of this approach is to protect connected objects and edge-enabled microdata centers against attackers. But the approach is not suitable in every aspect of the edge-enabling environment. Thus, for these reasons, blockchain Hyperledger-enabled distributed technology is proposed that provides distributed data transmission facilities along with data privacy, integrity, transparency, provenance, and availability (*Gill et al., 2024*; *Srirama, 2024*; *Zhang et al., 2024*). Therefore, it allows end-users to access the edge interface easily in a protected manner through the distributed application (DApp) (*Queralta & Westerlund, 2021*; *Al-Mamun et al., 2020*; *Firdaus, Rahmadika & Rhee, 2021*; *Palanisamy & Xu, 2024*). Most importantly, by using this, the system manages data encryption and preserves logs and transaction details on distributed storage (immutable storage) to enable integrity in the process of edge-based executions. The chain code controls the number of operations of edge devices autonomously in accordance with the current process of execution. By integrating these technologies, we can achieve secure, transparent, traceable, and protected preservation that is hard to alter, tamper with, or forge. Recently, in fog computing, the blockchain distributed technology has been envisioned and utilized by several giant enterprises to achieve integrity, confidentiality, transparency, and provenance because of the nature of the distributed network, which allows two-way channel transmission, such as on-chain (all the implicit chain transactions) and off-chain (all the explicit chain transactions).

## PROPOSED ARCHITECTURE

In this article, we present a proposed architecture by using blockchain Hyperledger enabling technology with an edge computing environment to integrate outsourced computation for the purpose of achieving secure data transmission, processing, and storage. The design architecture is divided into six different folds, such as (i) participating stakeholder registration, (ii) IoT, (iii) edge, (iv) fog, (v) cloud, and (vi) blockchain Hyperledger. First, we initiate end-user registration by collecting the request. After verification, the blockchain Hyperledger Sawtooth expert validates, creates a new registry, and exchanges updated information among the chains. Second, it is required to register all the IoT-enabling devices before initiating transactions. It is noted that the IoT devices have limited processing capabilities, as mentioned in the above section. However, each transaction is scheduled to move towards the edge because of the collected data processing. Edge captures data in accordance with the designed processes, such as (i) collection, (ii) processing, (iii) examination of critical aspects, (iv) analysis, and (v) presentation. This process reduces the cost of additional storage in both domains (static and dynamic). The edge gateway that is protected by the NuCypher re-encryption algorithm transmits analyzed records towards the fog environment for further outsourced computations where a lightweight computation is scheduled. The purpose of this setup is to reduce the computational cost of

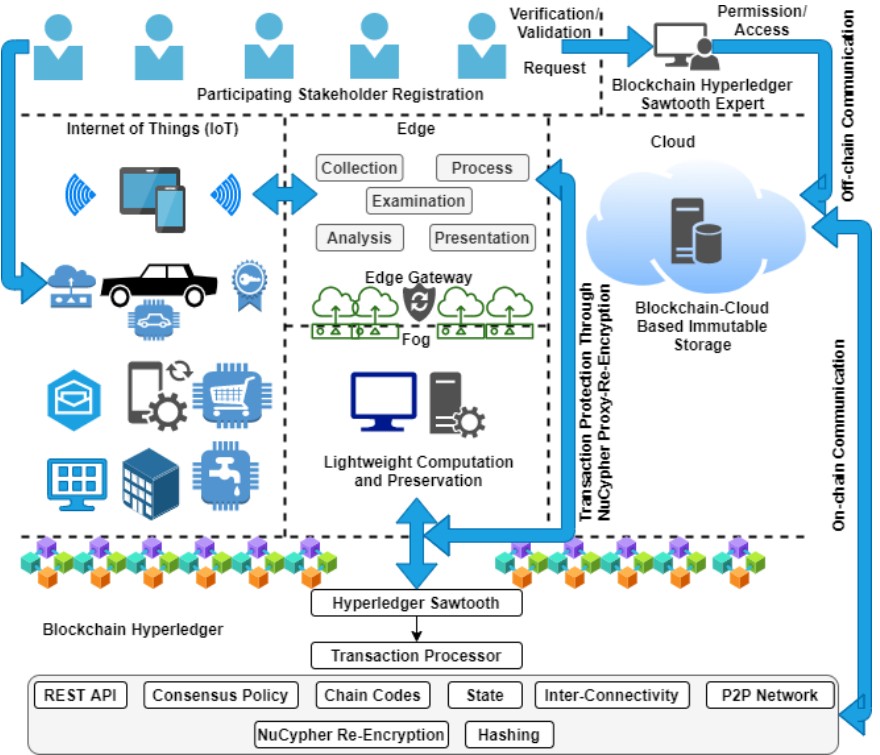

**Figure 5** Working sequences of the proposed work.

chunk-based data processing according to the priority bits, and therefore preserve all the logs (processing information, such as time of execution, delay, response, throughput, _etc._) of computations recorded in the fog storage for future analysis.

Fourth, cloud computing technology is integrated with edge, fog, and blockchain technologies for the purpose of reducing the computational, network, and storage load. Before preserving a record of every IoT transaction in the blockchain-cloud storage, it is examined and analyzed twice, as shown in Fig. 5. With the use of the NuCypher-proxy re-encryption algorithm, we protect ledger transactions in terms of the integrity and transparency of IoT data. All these processes are connected with the blockchain Hyperledger Sawtooth, which provides a secure, smooth, and protected event pathway for node transactions in the distributed environment (P2P network connectivity). However, by managing ledger privacy and security, the Sawtooth transaction processor is designed to handle the execution of the complete transaction occurring in the outsourced computational environment. In this scenario, the communication channel is split into two parts, such as on-chain and off-chain, where all the external transactions (explicit), for example, out of the outsourcing domain are handled by the off-chain communication channel. On the other side, all the internal transactions (implicit), for example, inside the

outsourced domain are handled by an on-chain communication channel, as shown in Fig. 5.

The remaining main blockchain Hyperledger Sawtooth-enabled prospects are used in the proposed architecture for secure data transmission in the integrated outsourced environment. The list of critical aspects highlighted in blockchain distributed ledger technology is discussed as follows:

- **Peer-to-peer network:** The distributed node connectivity between devices creates secure transmission channels where all the end-user's transactions are scheduled to be processed. One of the advantages of setting up is that it provides restricted direct deliverance of device messages because of path management (inter-communication (on-chain and off-chain)), as shown in Fig. 5. All the requests for data processing are received by outsourced nodes in a protected manner, such as integrity, confidentiality, provenance, security, and privacy, between a subspace of permissioned (private) and permissionless (public) network participants. However, the role of Hyperledger Sawtooth is critical, where it maintains edge-enabling distributed node services by means of transaction initiation, scheduling, organizing, managing, optimizing, preserving, and monitoring. Each transaction (protected with a hash-encryption mechanism) is endorsed after every participating stakeholder (permissioned only) signs digitally (a blockchain digital signature).

- **Certificate authority and protection:** In this proposed architecture, we designate certificate authority over the permissioned (private) and permissionless (public) networks to create trust between connected stakeholders. For this reason, the Hyperledger technology is considered one of the most protected environments as compared to other public chain-like structures (such as Ethereum), while data is shared among the connected participants in the chain.

- **Immutable preservation:** Blockchain-cloud-enabled immutable storage is designed to provide a secure data preservation option with a reduced cost of distributed storage as compared to other state-of-the-art storage options, such as Filecoin, IPFS, *etc.*

- **Chaincode, consensus, and digital signature:** For autonomous execution, we design chain codes, consensus policies, and digital signatures (as mentioned in Algorithm 1, Table 5). There are four main chain codes that are created that automate the execution of operations for end-user registration and participating stakeholders (endUReg()), adding new IoT-based transactions (listProcess(listProcess)), edge and fog-based computation and storage (updateChg()), and Blockchain-cloud-enabled record exchange (updateChg()).

## CURRENT STATE-OF-THE-ARTS

In this section, we discuss the current state of the art developments along with the open challenges, limitations, and issues involved in the classical edge-enabling distributed computing, and the difference when it collaborates with the blockchain Hyperledger-enabled and related integrations, and systematic evaluations fluctuations by connecting with advanced digital technologies.

**Algorithm 1.** Pseudo-implementation of chaincode and smart contracts.

Contract Initialization: Blockchain Experts for Outsourcing Computation are Responsible for handling Node-to-Node Transactions and Recording Addresses

Each Connected Node Passes Through the Process of Verification and Validation Before Joining

Data Received by the Outsource Node in a Chronological Order (Sequence Manner)

Data and Assumptions: For Initialization, It Requires Industrial Environment to Build Infrastructure

Data Generated by IoT Devices and Transmitted Through the Wireless Sensor Network

Outsource Received Data and Start the Process of Investigation or Execution

Variable Declaration and Initialization:

Open File: int main (): X.[a][file],

end user registration,

endUserReg; //variable declaration

IoT device registration,

deviceReg;

outsource node registration,

OutsourceNReg;

list of outsource process,

listProcess;

record data processing info,

recordDPI;

preserve ledger,

preLedger;

update changes,

updateChg;

Blockchain Hyperledger timestamp,

data(execution);

Blockchain Expert handles all the transactions and manages the overall run-time transmission and related executional procedures,

Record registration details and Addresses of endUserReg(endUserReg), deviceReg(deviceReg), OutsourceNReg(OutsourceNReg;) in the registration contract (endUReg(endUserReg)), and exchange info with the connected stakeholders;

Steps of Executions:

if endUserReg(endUserReg) ! = True

then, add user registration in endURre(endUserReg);

if data received and list of outsource process ! = True

then, process data according to the listProcess(listProcess) contract and record individually on updateChg(updateChg) and exchange;

additionally, record end user registration (endUserReg()), IoT device registration (deviceReg(deviceReg)), outsource node registration (OutsourceNReg(OutsourceNReg;)), list of outsource process (listProcess()), record data processing info (recordDPI()), preserve ledger (preLedger()), update changes (updateChg()), Blockchain Hyperledger timestamp data(execution) in accordance with the listProcess(listProcess) and addRec_ExcRec(), along with addresses;

request for changes occur from endUReg(), deviceReg(), OutsourceNReg(), add New Log (listProcess), the Blockchain Expert verifies and validates the request,

according to tuned Hyperledger-enabled consensus policy, transactions update share among the stakeholders using updateChg(updateChg);
else
check change state, remove error, add details, update, and exchange;
terminate;
else
check change state, remove error, add details, update, and exchange;
terminate;
Output: endUReg(), listProcess(), and updateChg();

## Interoperability, cross-platform connectivity, related connectivity between outsourced components and the future developments

Interoperability platforms (cross-platform connectivity) are considered one of the biggest challenges in the distributed network environment when two or more different chains exchange related information (*Khan et al., 2021b*). The concept of cross-chaining blockchain allows multiple connected nodes of different chains to share data without involving or participating in the particular chain. By developing and deploying the concept practically, the system is able to provide efficient and effective business services on the distributed network (*Nguyen et al., 2021*; *Shaikh et al., 2022b*). However, the infrastructure of the cross-chain platform provides a distributed application (DApp) environment, where a secure and protected channel is designed that handles a number of internal and external transactions within and outside of the chain (*Vashishth et al., 2024*; *Tanwar et al., 2024*). The end users send data for outsourcing computation purposes to the fog node through DApp, interact with different connected nodes, and preserve processed records (logs) in the immutable storage. All the records are stored in this distributed storage in chronological order with a chain-like structure (*Ahuja & Deval, 2021*; *Khanagha et al., 2022*; *Matrouk & Alatoun, 2021*; *Muniswamaiah, Agerwala & Tappert, 2021*; *Sabireen & Neelanarayanan, 2021*; *Singhal & Singhal, 2021*).

The development of a cross-chain platform for managing outsourcing nodes connectivity, transmission, and storage to improve the processes of data receiving, the layered hierarchy of data transmission, maintain resource consumption, and securely preserve information. Therefore, it conducts meaningful and secure chain-to-chain transactions at different connected outsource nodes. However, the exiting fog-enabled computational legacy and service delivery architecture and its associative connectivity with distributed components create a lack of platform cross-chaining. Until now, it has been hard to adopt and deploy such a type of platform because of the current immature infrastructure and due to nodes' disunity and fragile interconnectivity.

## Scope of scalability and privacy protection and preservation

The existing fog-enabled outsourcing environment is facing a serious challenge while connected with the blockchain Hyperledger technology; implementation of a blockchain network and initiating transactions in a distributed environment are two of the biggest issues because it requires cost (guest and host fees) to schedule nodes' data exchange in a secure

**Table 5  Pseudo-implementation of the working hierarchy of the proposed framework.**

**Pseudo-implementation of chaincode and smart contracts.**

Contract Initialization: Blockchain Experts for Outsourcing Computation are Responsible for handling Node-to-Node Transactions and Recording Addresses
Each Connected Node Passes Through the Process of Verification and Validation Before Joining
Data Received by the Outsource Node in a Chronological Order (Sequence Manner)
Data and Assumptions: For Initialization, It Requires Industrial Environment to Build Infrastructure
Data Generated by IoT Devices and Transmitted Through the Wireless Sensor Network
Outsource Received Data and Start the Process of Investigation or Execution
Variable Declaration and Initialization:
Open File: int main (): X.[a][file],
end user registration,
endUserReg; //variable declaration
IoT device registration,
deviceReg;
outsource node registration,
OutsourceNReg;
list of outsource process,
listProcess;
record data processing info,
recordDPI;
preserve ledger,
preLedger;
update changes,
updateChg;
Blockchain Hyperledger timestamp,
data(execution);
Blockchain Expert handles all the transactions and manages the overall run-time transmission and related executional procedures,
Record registration details and Addresses of endUserReg(endUserReg), deviceReg(deviceReg), OutsourceNReg(OutsourceNReg;) in the registration contract (endUReg(endUserReg)), and exchange info with the connected stakeholders;
Steps of Executions:
if endUserReg(endUserReg) != True
then, add user registration in endURre(endUserReg);
if data received and list of outsource process != True
then, process data according to the listProcess(listProcess) contract and record individually on updateChg(updateChg) and exchange;
additionally, record end user registration (endUserReg()), IoT device registration (deviceReg(deviceReg)), outsource node registration (OutsourceNReg(OutsourceNReg;)), list of outsource process (listProcess()), record data processing info (recordDPI()), preserve ledger (preLedger()), update changes (updateChg()), Blockchain Hyperledger timestamp data(execution) in accordance with the listProcess(listProcess) and addRec_ExcRec(), along with addresses;
request for changes occur from endUReg(), deviceReg(), OutsourceNReg(), add New Log (listProcess), the Blockchain Expert verifies and validates the request, according to tuned Hyperledger-enabled consensus policy, transactions update share among the stakeholders using updateChg(updateChg);
else
check change state, remove error, add details, update, and exchange;
terminate;
else
check change state, remove error, add details, update, and exchange;
terminate;
Output: endUReg(), listProcess(), and updateChg();

manner. Further, it also requires additional charges to maintain the security scalability and efficiency of IoT-enabled chains of transactions while the size of node transactions fluctuates (*Shaikh et al., 2022b*). However, Hyperledger's technologically enabled modular architecture achieves integrity, traceability, provenance, and trustworthiness by stimulating the digital ledger in a distributed network environment by executing the node transactions and incorporating execution details with the end users and stakeholders. In this manner, the connected end users can see the operational executions and related details of the transactions regardless of whether they send a request to the manager of the ecosystem or wait for managerial approval (*Laghari et al., 2023*; *Varadam et al., 2024*). Undoubtedly, Hyperledger provides several of these types of advantages, but the cost of guest fees for events of node transactions decreases the rate of stakeholders' adaptation. The authority of Hyperledger technology needs to collaborate with Ganache (a platform that provides free test accounts). It will allow developers (because of the free hosting) to check the design and implementation of the proposed blockchain-enabled outsourcing computational architecture and its related events of node transaction executions before deploying the original one.

## Data management and optimization distribution

In the current cloud-enabled outsourcing computation and resource management environment, the only available option is to store chain-of-records in client–server-based centralized storage (*Khan et al., 2021b*; *Nguyen et al., 2021*; *Shaikh et al., 2022b*; *Vashishth et al., 2024*; *Tanwar et al., 2024*; *Laghari et al., 2023*; *Varadam et al., 2024*). This type of data management and central storage leads to the ledger being compromised when malicious attacks are attempted through the network, such as distributed denial of service (DDoS). Further, there is also no proper structure presented that blocks internet attacks and creates a susceptible environment to prevent information integrity while transmitting (*Khan et al., 2021a*; *Nikravan & Kashani, 2022*; *Datta & Namasudra, 2024*; *Li et al., 2024*; *Sinha, Singh & Verma, 2024*). However, the use of blockchain distributed ledger technology with a hash-encryption cryptographic algorithm to protect individual transmissions and store the details of logs generated in immutable storage. In the proposed blockchain-cloud storage, a third-party data storage structure is used that charges a reduced cost (less than 10 dollars per month) for data preservation and prevention. It connects the Hyperledger modular infrastructure and alleviates data management and privacy-related issues by providing data integrity, transparency, traceability, and provenance. Therefore, the property directly associated with blockchain, and Hyperledger-enabling technologies allows the system to protect itself from intrusions. The deployment of a distributed application (DApp) that runs chain code-enabled solutions to automate outsourcing computations and related integration and preservation of the processed record in a distributed cloud environment and turn the transparency of the ledger.

## Role of regulatory management, compliances and protocols for making standardization

Within the outsourcing computational environment, a vast and diverse range of data received needs processing concurrently (*He et al., 2024*). This complete procedure

contributes to the lack of regulatory compliance and standardization because there is no standard process model presented previously and no secure layered hierarchy proposed (*Alamer, 2024*; *Sasikumar et al., 2024*). The traditional processed hierarchy of data receiving, examining, analyzing, storing, presenting, and reporting in the outsource node is less reliable and unsecure (*Ajani et al., 2024*; *Cui et al., 2024*; *Chen et al., 2024*; *Mehmood et al., 2024*; *Ramya & Ramamoorthy, 2024*; *Bibri et al., 2024*; *Rani & Srivastava, 2024*). In the results, an unavoidable negative impact leads to inconsistent executions that create consequences in terms of quality. The whole scenario will improve by enabling the blockchain technology to be integrated with the fog nodes that collaboratively design a novel process in accordance with cross-boarding standardization, where Hyperledger modular infrastructure enforces the standard procedure that automatically enhances the quality of data executions.

## CONCLUSIONS

This article discusses the emerging issues and limitations of the current integration strategies of edge computing with outsourcing computation. In this scenario, the number of records (generated by IoT devices) is processed individually in a distributed manner. In the industrial environment, this process helps to solve various multi-execution-related problems, enhance productivity, and reduce the consumption of cloud-enabled storage. But for all these benefits, there are a few gaps that need attention, such as privacy and security issues in the data processing, sharing, exchanging, and preserving transactional ledgers during the complete cycle of transmission. The evaluation of blockchain enables technology to provide a secure channel for data transmission, including the exchange of processed information among stakeholders and its preservation. It only allows participating stakeholders (within the chain) to schedule on-chain and off-chain transactions, share data, and exchange related details in the distributed P2P consortium network. However, the involvement of blockchain Hyperledger technology with edge computing led to the resolution of several privacy issues while integrated with the outsourced (fog) computations, as mentioned in Tables 1 and 2 (systematic analysis). Therefore, in this article, we propose a blockchain Hyperledger Sawtooth-enabled, novel, and secure architecture for edge-fog-based outsourcing computation integration, along with a consortium channel for secure data transmission. Further, the proposed architecture provides on-chain and off-chain communication protocols that execute the events of node transactions implicitly and explicitly, respectively. In addition, the associative NuCypher re-encryption algorithm manages individual transaction protection so that the delivery of every transaction occurs securely while preserving the details of the complete execution of the overall data initiated (captured) and delivery process. To automate the executions, we designed a pseudo-chain of smart contracts and consensus policies that handle and manage the node (edge and fog) registration, record new transactions, manage data, process individual entities, organize

and optimize records, and update the ledger. By analyzing these benefits, we can say that the proposed architecture will become a good candidate for real-time industrial development.

### Funding
The authors received no funding for this work.

### Competing Interests
Natalia Kryvinska is an Academic Editor for PeerJ. The authors declare there are no competing interests.

### Author Contributions
- Abdullah Ayub Khan conceived and designed the experiments, performed the experiments, analyzed the data, performed the computation work, prepared figures and/or tables, authored or reviewed drafts of the article, and approved the final draft.
- Asif Ali Laghari performed the experiments, authored or reviewed drafts of the article, and approved the final draft.
- Abdullah M. Baqasah performed the computation work, authored or reviewed drafts of the article, and approved the final draft.
- Roobaea Alroobaea analyzed the data, authored or reviewed drafts of the article, and approved the final draft.
- Ahmad Almadhor conceived and designed the experiments, performed the computation work, prepared figures and/or tables, authored or reviewed drafts of the article, and approved the final draft.
- Gabriel Avelino Sampedro performed the experiments, prepared figures and/or tables, authored or reviewed drafts of the article, and approved the final draft.
- Natalia Kryvinska performed the experiments, analyzed the data, performed the computation work, authored or reviewed drafts of the article, and approved the final draft.

### Data Availability
 This is a literature review.

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
