# Peer review of "Blockchain-enabled infrastructural security solution for serverless consortium fog and edge computing"

_PeerJ Computer Science, doi:10.7717/peerj-cs.1933_

## Round 0.1 · original submission · Minor Revisions

Reviewer 1 raises concerns about the paper's organizational clarity due to inconsistent section numbering and the lack of detailed explanations for tables summarizing previous research. They also suggest minor improvements, including providing specific numbers in the literature review and maintaining consistent tense usage. In contrast, Reviewer 2 commends the paper's organization and comprehensive literature coverage but notes format issues and the need to improve table readability. Both reviewers acknowledge the importance of the open research questions presented in the paper, highlighting its value to the community.

**Language Note:** PeerJ staff have identified that the English language needs to be improved. When you prepare your next revision, please either (i) have a colleague who is proficient in English and familiar with the subject matter review your manuscript, or (ii) contact a professional editing service to review your manuscript. PeerJ can provide language editing services - you can contact us at copyediting@peerj.com for pricing (be sure to provide your manuscript number and title). – PeerJ Staff

Reviewer 1 ·

Basic reporting

This paper presents a systematic study of blockchain Hyperledger enabling fog and edge computing, which addresses an important problem nowadays. This paper studies previous research in edge distributed computing and outsourcing computation and proposes a blockchain Hyperledger sawtooth-enabled architecture for edge-fog-based outsourcing computation integration, emphasizing secure data transmission and privacy concerns. Also, the authors propose some open research problems for future research. However, there are several parts can be improved.

Experimental design

In my review of the paper, I noticed an inconsistency regarding section numbering. The authors refer to sections 2-7 in lines 187-198, yet the paper itself does not appear to have these sections clearly numbered. This lack of clear section demarcation makes it challenging to follow the paper's organizational logic and structure.

Validity of the findings

One main issue is about the study of previous paper. The authors give two tables summarizing the status and open issues of existing research, but don't give full explanations on the two tables. This makes the readers hard to understand the content of the tables and the conclusions the authors got from their studies. I would recommend the authors explain more details about the table contents, and how the authors analysis the results and get useful conclusions.

Additional comments

Some minor issues:
1. In literature review articles, it is better to include specific numbers, like " by using xxx techniques, how many papers are found, ..."

2. There is an inconsistency in tense usage between lines 174 and 180. In academic and technical writing, maintaining a consistent tense is crucial for clarity and coherence.

3. At line 144, the use of the word "assume" could potentially convey a sense of uncertainty or lack of confidence in the authors' conclusions. It might be better to use other words.

Reviewer 2 ·

Basic reporting

This paper provided a systematic literature review using blockchain-enabled technologies to handle security and privacy issues in IOT/edge devices environment.

The authors first provided reviews on edge computing, existing security and privacy issues, and the current research of using blockchain to protect integrity. They then provided an architecture and highlight some open research questions.

It's commendable that the authors made distinction of this review with the previous literature reviews in table 2.

Experimental design

The review is very organized into different sections and included enough amount of literatures.

Validity of the findings

The authors provided interesting open research questions as the findings of summarizing existing works.
They discussed issues such as interoperability, cross-platform connectivity, scalability, privacy concerns; the need for a cross-chain platform to facilitate efficient data exchange; challenges related to transaction costs and data management; and highlight the lack of compliance, regulatory management, and standardization. These are interesting research topics for the community to further dig into.

Additional comments

Thanks for submitting your work. I find it includes enough background information and also surveyed enough existing research works. I have several presentation issues for the authors to address.

[Format checking] Please ensure that the paper if formatted correctly. There are several format issues in this paper. For example, on the first page, line 71 and 72 contains empty spaces before each line.

[move figures and tables] tables and figures are included in the end of the paper, please move them to the corresponding place in the main paper. Otherwise, it's very hard for the readers to follow the flow.

[improve table 2] please consider adding lines between each column in table 2.

[fix table 5] on the last page, there are many empty lines, please address them correctly.

---

## Round 0.2 · Major Revisions

One Section Editor has commented and said:

"The paper is not up to date. The latest reference is from 2021. This is a survey paper. However, it does not discuss the current state of the art. So, I don't recommend acceptance with this presentation."

Please include additional recent papers (e.g., 30 more up-to-date citations), and compare them in depth.

Reviewer 1 ·

Basic reporting

The authors address most comments quite well. I only have some minor suggestions: put some content of Table 2 and Table 3 into the text sections.

Experimental design

no comment

Validity of the findings

no comment

Additional comments

no comment

Reviewer 2 ·

Basic reporting

N/A

Experimental design

N/A

Validity of the findings

N/A

Additional comments

Thanks for submitting the revised version. I appreciate the authors' efforts of making the changes. It clears most of the concerns from my previous review.

Though the figures are still in the end of the paper, and I personally would prefer it to be integrated into the original places in the main paper instead of putting things to the end. But the authors also mentioned that this follows the journal's formatting style. So given this case, I would be fine for the paper to be accepted in the current format.

---

## Round 0.3 · accepted · Accept

The authors have added up-to-date citations, and therefore, I think we should accept the paper.